# Optimisation of Urine Sample Preparation for Headspace-Solid Phase Microextraction Gas Chromatography-Mass Spectrometry: Altering Sample pH, Sulphuric Acid Concentration and Phase Ratio

**DOI:** 10.3390/metabo10120482

**Published:** 2020-11-25

**Authors:** Prashant Aggarwal, James Baker, Mark T. Boyd, Séamus Coyle, Chris Probert, Elinor A. Chapman

**Affiliations:** 1Department of Molecular and Clinical Cancer Medicine, Institute of Systems, Molecular and Integrative Biology, University of Liverpool, Liverpool L69 3BX, UK; P.Aggarwal3@student.liverpool.ac.uk (P.A.); hljbake3@liverpool.ac.uk (J.B.); chris.probert@liverpool.ac.uk (C.P.); 2School of Medicine, Cedar House, University of Liverpool, Liverpool L69 3GE, UK; 3Department of Molecular and Clinical Cancer Medicine, Institute of Systems, Molecular and Integrative Biology, Cancer Research Centre, University of Liverpool, Liverpool L3 9TA, UK; mboyd@liverpool.ac.uk; 4Palliative Care Institute Liverpool, Cancer Research Centre, University of Liverpool, Liverpool L3 9TA, UK; S.Coyle@liverpool.ac.uk; 5Clatterbridge Cancer Centre, Liverpool L7 8YA, UK; 6School of Medical Sciences, Bangor University, Bangor, Gwynedd LL57 2DG, UK

**Keywords:** volatile organic compounds, VOCs, H_2_SO_4_, NaOH, HCl, sodium hydroxide, hydrochloric acid, HS-SPME-GC-MS, vials

## Abstract

Headspace-solid phase microextraction gas chromatography-mass spectrometry (HS-SPME-GC-MS) can be used to measure volatile organic compounds (VOCs) in human urine. However, there is no widely adopted standardised protocol for the preparation of urine samples for analysis resulting in an inability to compare studies reliably between laboratories. This paper investigated the effect of altering urine sample pH, volume, and vial size for optimising detection of VOCs when using HS-SPME-GC-MS. This is the first, direct comparison of H_2_SO_4_, HCl, and NaOH as treatment techniques prior to HS-SPME-GC-MS analysis. Altering urine sample pH indicates that H_2_SO_4_ is more effective at optimising detection of VOCs than HCl or NaOH. H_2_SO_4_ resulted in a significantly larger mean number of VOCs being identified per sample (on average, 33.5 VOCs to 24.3 in HCl or 12.2 in NaOH treated urine) and more unique VOCs, produced a more diverse range of classes of VOCs, and led to less HS-SPME-GC-MS degradation. We propose that adding 0.2 mL of 2.5 M H_2_SO_4_ to 1 mL of urine within a 10 mL headspace vial is the optimal sample preparation prior to HS-SPME-GC-MS analysis. We hope the use of our optimised method for urinary HS-SPME-GC-MS analysis will enhance our understanding of human disease and bolster metabolic biomarker identification.

## 1. Introduction

### 1.1. Metabolomics and Volatile Organic Compounds

Metabolomics identifies de novo or changing metabolites, often in the form of volatile organic compounds (VOCs) from biological samples [1]. In the 1970s, Horning et al. used gas chromatography-mass spectrometry (GC-MS) to measure VOCs in human urine and tissues for the first time [2]. Since then, the study of VOCs has grown exponentially [3,4]. However, there is no widely adopted standardised protocol for the preparation of urine samples for analysis resulting in an inability to compare studies reliably between laboratories [5,6].

VOCs usually have fewer than 12 carbon atoms, a boiling point of less than 300 °C, and are biological or synthetic in origin [3,5,6,7]. They have a high vapour pressure and low molecular weight enabling them to enter the gas phase at ambient temperatures; they may have an odour and emanate from urine, faeces, saliva, breath, and other bodily products [5,8]. VOCs are produced by a variety of processes including degradation from metabolic pathway intermediates, and, therefore, their concentration in samples can give an insight into biochemical activity upstream, with potential to be biomarkers of disease [2,6].

So far, VOC analysis of faeces has shown: (i) the increase in esters of short chain fatty acids, cyclohexanecarboxylic acid and its ester derivatives associated with inflammatory bowel syndrome (diarrhoea type) [9]; (ii) the loss of short chain fatty acid in active inflammatory bowel disease [10,11] and, finally, (iii) an increase in propan-2-ol and the ratio of propan-2-ol to 3-methylbutanoic acid in colorectal cancer [12]. Urinary VOCs have unsurprisingly been proposed to be useful to detect metabolic changes in conditions involved in urological systems, for example, urinary tract infection [13], minimal change type nephrotic syndrome [14], and urological cancer detection (kidney, renal cell carcinoma, and bladder) [1,15,16,17]. Moreover, urinary VOCs have also been proposed for the detection of a wide range of other cancers outside the urological systems [18,19], including colorectal cancer [20], head and neck cancer [21], and lung cancer [22]).

Regarding the classes of VOCs detected in biological materials, De Lacy Costello et al. reviewed VOCs present in different biological materials and found that the proportion of compound classes varied between biological materials. Urine contained a high number of ketones, furans, and ethers and almost no ester compounds [23]. More recently, a review on lung cancer urinary VOCs suggested that ketones, aldehydes, and ethyl acetate could be useful in detecting lung cancer patients [22]. Excitingly, the use of VOCs for diagnosis is still in its infancy.

### 1.2. Analysis of Volatile Organic Compounds

Detection of VOCs requires precise, reliable, and effective instrumentation [23,24,25,26,27]. One of the more popular techniques is headspace-solid phase microextraction coupled with GC-MS (HS-SPME-GC-MS) [4,7,8,23,28]. It is a fast and cost-efficient technique that has identified substantial numbers of VOCs in biological material from both humans and other species [5]. HS-SPME-GC-MS delivers an exceptional combination of metabolite selectivity and specificity [29]. The solid phase microextraction (SPME) fibre adsorbs VOCs in the headspace of a sample—improving on GC-MS alone. It provides a small, low-cost, and solvent-free pre-concentration technology which simplifies sample preparation into one step [6,8,30].

### 1.3. Why Urine?

A variety of biofluids are amenable to VOC analysis by HS-SPME-GC-MS; however, urine is ideal as it can be sampled frequently, easily, non-invasively, and stored for long periods [31]. Additionally, urine samples generate an intricate and comprehensive metabolomic profile, with VOCs ranging in polarity and complexity [4]. The added benefit of using urine is that it contains end-products from multiple metabolic processes, unlike faeces which predominantly represents the alterations in the gut. Breath is a potential alternative to urine, as it is easy to collect and represents a ‘filtrate’ of blood passing though the lungs. However, breath is difficult to store and the concentrations of VOCS are low [32,33,34]. Urine appears to be an ideal biofluid for biomarker screening, clinical diagnoses, and the monitoring of treatment [1,21,25,35,36,37].

### 1.4. Sample Preparation

Different sample preparation methods optimise VOC detection including adding salt, acid or base to the sample [2,6,7,8,38]; freeze-drying (removing water from a sample while keeping it frozen) [6]; and altering phase ratio (the ratio between sample volume and gas volume in a vial), by changing the volume of the vial or the urine sample [6,7,39]. Acidified treatment techniques produce more VOCs than neutral pH and the Divinylbenzene/Carboxen/Polydimethylsiloxane (DVB/CAR/PDMS) SPME fibre enhances analytic performance more than other types of SPME fibres [5,39]. However, the current evidence base is small. The above methods change the ionic strength, polarity or solubility of the samples resulting in a higher concentration of VOCs in the gaseous phase.

Sulphuric acid as a sample preparation treatment technique has rarely been explored and information altering phase ratio is inconclusive [5,8,40].

This paper investigates the effect of altering urine sample volume, vial size, and pH for optimising detection of VOCs when using HS-SPME-GC-MS.

## 2. Results

### 2.1. Altering Urine Sample pH by the Addition of 5 M H_2_SO_4_, 5 M HCl, or 5 M NaOH

#### 2.1.1. Number of VOCs

Urine samples’ (1 mL) pH was altered by the addition of 0.2 mL 5 M H_2_SO_4_, 5 M HCl, or 5 M NaOH (to generate sample solutions of 0.83 M, pH 0.075, pH 0.081, and pH 13.919, respectively). The mean number of VOCs per sample for each treatment technique (±standard deviation (SD), coefficient of variation (CV)) were 33.5 (±10.3, 30.7%), 24.3 (±9.6, 39.5%), and 12.2 (±3.0, 24.6%) in H_2_SO_4_, HCl, and NaOH, respectively; all differences were significant (*p* < 0.01; Wilcoxon signed-rank test) (*n* = 26) (Figure 1A). There were 73 unique VOCs in the H_2_SO_4_ treatment group compared to 16 in NaOH (Figure 1B). Evaluating the 31 VOCs shared between the H_2_SO_4_ and NaOH treated samples, three VOCs in H_2_SO_4_ (2-buta-1,3-dienyl-1,3,5-trimethylbenzene, methyldisulfanylmethane, and hexanal) and two VOCs (heptan-2-one and pentan-2-one) in NaOH had significantly greater abundances compared to their counterparts (*p* < 0.05, respectively; FC > 1.5, FC < −1.5). There were 27 unique VOCs in the H_2_SO_4_ treatment group compared to four in HCl (Figure 1C). Evaluating the 77 VOCs shared between both H_2_SO_4_ treatment and HCl treatment groups, no compounds had a significantly greater abundance compared to their counterpart (*p* < 0.05, respectively; FC > 1.5, FC < −1.5).

#### 2.1.2. Classification of VOCs

5 M H_2_SO_4_, 5 M HCl, and 5 M NaOH treatment groups (sample solutions of 0.83 M; pH 0.075, pH 0.081, and pH 13.919, respectively) had a total of 18, 16, and 15 different compound classes. The 5 M H_2_SO_4_ treatment group (0.83 M final concentration, pH 0.075) produced more classes of VOCs than 5 M NaOH (0.83 M final concentration, pH 13.919) (Figure 1D). H_2_SO_4_ treatment identified a higher average number of VOCs classified as aldehyde, alkaloid, azetidine, benzene derivative, carboxylic acid, cosmetic, cycloparaffin, furan, sulphur compound, terpene or unclassified; while it identified a lower average number of VOCs classified as alcohol, alkane, amine, azole, contaminant, ketone, tetrazine, therapeutic, and unknown. Compared to HCl (pH 0.081), the H_2_SO_4_ (pH 0.075) treatment group resulted in a higher average number of VOCs classified as aldehyde, alkaloid, azole, carboxylic acid, furan, ketone, and sulphur compounds; while it identified a lower average number of VOCs classified as alcohol, amine, azetidine, benzene derivative, contaminant, cosmetic, cycloparaffin, terpene, tetrazine, unclassified, and unknown (Figure 1D). The full breakdown of classification of VOCs can be found in Appendix A, and per treatment group in Appendix A.

#### 2.1.3. HS-SPME-GC-MS Degradation

The H_2_SO_4_ treatment group (sample solutions of 0.83 M, pH 0.075) was associated with less degradation than with either 5 M HCl (sample solutions of 0.83 M, pH 0.081) or 5 M NaOH (sample solutions of 0.83 M, pH 13.919). There were five compounds of degradation that were more abundant with HCl than H_2_SO_4_ and just one that was more common with H_2_SO_4_ (*p* < 0.01, respectively; Wilcoxon-signed rank test). NaOH was associated with five degradation compounds that were significantly more abundant than compared to H_2_SO_4_ (*p* < 0.01, respectively; Wilcoxon-signed rank test) (Figure 2, Table 1). These degradation products could be coming from either column degradation, the fibre, vial lid or the septa of the GC inlet.

#### 2.1.4. Conclusion on Altering Sample pH

Overall, altering sample pH using H_2_SO_4_ (final concentration 0.83 M, pH 0.075) resulted in more VOCs being detected than following either of the other treatments. H_2_SO_4_ treatment resulted in a significantly larger mean number of VOCs per sample and more unique VOCs, produced more VOCs with significantly greater abundances than their counterparts, produced a more diverse range of classes of VOCs, and led to less HS-SPME-GC-MS degradation products being detected.

### 2.2. Altering Concentration of H_2_SO_4_

The mean (±SD, CV) numbers of VOCs per sample resuspended with 5 M, 2.5 M, and 1 M H_2_SO_4_ (to generate sample solutions of 0.83 M, 0.42 M, and 0.17 M, respectively, pH 0.075, pH 0.365, and pH 0.743, respectively) were 30.3 (±6.5, 21.5%), 24.2 (±6.7, 27.7%), and 16.1 (±7.9, 49.1%), respectively (*n* = 15). The 1 M concentration has a significantly lower mean number of VOCs than 2.5 M and 5 M treated samples, respectively (*p* < 0.05 and *p* < 0.01; Wilcoxon signed-rank test); however, there was no statistically significant difference in the mean number of VOCs between 2.5 M and 5 M concentrations. There were 4 unique VOCs in the 2.5 M concentration compared to 12 VOCs in 5 M (Figure 3). Evaluation of the 71 VOCs found commonly when samples are resuspended in either concentration of H_2_SO_4_ showed no significant differences in the abundances of VOCs produced (*p* < 0.05; FC > 1.5, FC < −1.5).

### 2.3. Altering Phase Ratio of H_2_SO_4_ Treated Urine Samples via Altering Vial Volume and Volume of Urine

#### 2.3.1. The Vial Size

The mean (±SD, CV) numbers of VOCs per 0.5 mL sample (treated with 0.1 mL 5 M H_2_SO_4_, to yield 0.83 M final solution, pH 0.075) for each vial size were 20.8 (±9.3, 44.7%) and 27.6 (±9.7, 35.1%) in 2 mL and 10 mL vials, respectively; the difference was significant (*p* < 0.01; Wilcoxon signed-rank test) (*n* = 15). There were 14 unique VOCs in the 10 mL vial compared to 3 in the 2 mL vial (Figure 4). Evaluating the 62 VOCs shared between the vial sizes showed that one VOC ((methyldisulfanyl)methane) in the 2 mL had a significantly greater abundance compared to its counterpart (*p* < 0.05; FC > 1.5, FC < −1.5).

#### 2.3.2. The Volume of Urine

The mean (±SD, RSD/CV) numbers of VOCs per sample for each volume of urine (treated with 5 M H_2_SO_4_, to yield 0.83 M final solution, pH 0.075) were 30.5 (±10.5, 34.4%) and 27.6 (±9.7, 35.1%) in 1 mL and 0.5 mL of urine, respectively; the difference was significant (*p* < 0.01; Wilcoxon signed-rank test) (*n* = 15). There were 17 unique VOCs in the 1 mL urine sample compared to six VOCs in the 0.5 mL urine sample (Figure 5). Evaluating the 70 VOCs shared between volume of urine samples, showed no significant differences in the abundances of VOCs produced (*p* < 0.05; FC > 1.5, FC < −1.5).

## 3. Discussion

HS-SPME-GC-MS analysis of urine offers the potential for fast, non-invasive metabolite detection but standardisation of methods between laboratories remains a challenge. We have shown that altering urine sample pH via the addition of H_2_SO_4_ is more effective at optimising detection of VOCs than NaOH or HCl. Ionic strength due to the addition of saturated NaCl solution was also examined in pooled QC samples and sample urine. We found less compounds were detected when using saturated NaCl compared to any other method of preparation (Appendix A). H_2_SO_4_ resulted in a significantly larger mean number of VOCs identified per sample and more unique VOCs, produced a more diverse range of classes of VOCs, and led to less HS-SPME-GC-MS degradation. In addition, we found that resuspension of samples with 2.5 M and 5 M H_2_SO_4_ yielded significantly better detection of VOCs compared to 1 M acid. Furthermore, using a 10 mL vial and 2.5 M or 5 M H_2_SO_4_ further optimised VOC detection.

Different urine sample treatment techniques alter the sample matrix. Firstly, change in pH may lead to an increase in decomposition or degradation of certain compounds present in the sample. In particular, using strong oxidizing acid will almost certainly change the sample composition. Secondly, changing either the ionic strength or pH may increase activity coefficients (how much a solution differs from an ideal solution), in turn, decreasing the partition coefficients (the ratio of the concentration of a sample in the liquid phase with the concentration of VOCs in the headspace), and, ultimately, resulting in more compounds transitioning into the gas phase. It known that an acid environment leads to a larger number of VOCs produced compared with a neutral or alkaline environment [6,38]. Despite this, there is a lack of consistency in the approaches described in the literature with either acid, alkali, and/or salts used as a sample treatment [2,8,16,31,41,42,43,44,45,46].

When acidification is chosen for HS-SPME-GC-MS, HCl appears to be the most common choice; although, the reason for this is typically not explained. We found here that HCl produced more degradation products than H_2_SO_4_. Neither HCl nor H_2_SO_4_ is very volatile and therefore they can accumulate at the front of the column. If they remain there, they will damage the stationary phase of the HS-SPME-GC-MS resulting in premature and excessive column bleed, reflected by an increased detection of degradation products. Both acids may lead to some deterioration of the SPME fibre; this requires further study. HCl is a stronger acid than H_2_SO_4_, which may explain the greater number of compounds of degradation. Furthermore, samples resuspended with HCl yielded a lower number of VOCs than H_2_SO_4_ from our urine samples. Notably, the only other paper that has described the use of H_2_SO_4_ as a urine sample treatment technique for HS-SPME-GC-MS analysis, did not compare different acidification techniques for optimising urinary VOCs in HS-SPME-GC-MS [8]. After establishing H_2_SO_4_ as the most effective treatment technique, we conducted further experiments altering other aspects of sample preparation, such as H_2_SO_4_ concentration and phase ratio.

A higher concentration of acid will lead to a greater increase in the activity coefficients, resulting in a larger concentration of compounds in the headspace. Previously, 1 M H_2_SO_4_ has been used; however, we found that both 2.5 M and 5 M H_2_SO_4_ were significantly more effective at producing VOCs detected using our HS-SPME-GC-MS method [8]. There was no significant difference in the number of VOCs produced between 5 M and 2.5 M H_2_SO_4_; the reason for this is likely to be that the SPME fibre is saturated with VOCs at 2.5 M, and, thus, there was no difference when using a higher concentration [5]. We anticipate using a more concentrated acid may lead to more extensive damage to the column or SPME fibre, leading to further contamination of the samples. Future studies evaluating the effects of changing the volume of acid used at the same concentration would be beneficial. A recent report recommended the use of 1 M H_2_SO_4_ for the optimisation of liquid-liquid extraction (LLE) coupled with GC-MS. This is an alternative to SPME but it is more tedious, time-consuming, and requires larger volumes of solvents [47,48,49].

Over the course of our study the repeated use of pooled urine QC samples at regular intervals indicated that there was no significant problem with volatile compound detection, suggesting adsorption was not being decreased over this period of time (Figure 6). Further, looking at the three most abundant compounds over time in a larger experiment using 5 M H_2_SO_4_, over a period of 15 days, where we ran over 200 samples, indicated that there was no change to detection over this time period (Appendix A). As sulphuric acid is a strong oxidizing acid, its use will lead to the formation of sulphates, and whether these enter the gas phase and lead to fibre damage requires further study. Future studies should also include a study of the long-term stability of the fibre under acidic conditions. This study would physically measure the fibre and continue to use pooled QC samples in order to show that the fibre was intact and continuing to adsorb efficiently. This study would indicate the optimal time to change the fibre. These further studies on SPME fibre degradation are vital, since SPME fibres have a high cost associated with them. It is therefore important to consider how many samples can be run per fibre to make this a viable avenue moving forward.

The phase ratio depends upon the sample volume and the vial size. Theoretically, an increased sample volume and a decreased vial size should result in more compounds in the headspace due to an increased phase ratio. However, the degree of change is dependent on the compounds’ partition coefficients [50]. If the partition coefficient of a compound is low (prefers the headspace phase) then altering the phase ratio will significantly increase the concentration of a compound in the headspace, and vice versa. Therefore, increasing urine volume results in a larger concentration of VOCs in the headspace, as exemplified by one of the compounds we identified: *n*-hexane, which is known to have a low partition coefficient, and is present in more of the 1 mL compared to the 0.5 mL urine samples. 

Previous research found altering vial volume had no significance [5,40]. The two vial sizes we selected are the most commonly used for HS-SPME analysis. We found the 10 mL vial was more effective than the 2 mL vial. It produced a significantly larger mean number of VOCs per sample and more unique VOCs. We propose that a larger headspace (as with the 10 mL vial) creates a larger surface area, and, thus, a greater opportunity for diffusion of gases into the headspace [50]. We hypothesise that because the 10 mL vial has a greater diameter than the 2 mL vial, the greater surface area enables more compounds to transition into the headspace from the liquid phase [50].

This study benefits from the use of duplicate or triplicate urine samples, to compare three simple treatment techniques (H_2_SO_4_, HCl, and NaOH). Our sample preparation is straightforward. Once a treatment solution is ready for use, it can quickly and easily be added to urine samples prior to HS-SPME-GC-MS analysis [6]. We believe analysis is further enhanced by randomising sample analysis, as this minimises the bias introduced when preparing and analysing replicate samples [51].

Internal standards were not used in this study because they are not practical for untargeted metabolomics [52,53,54]. Our QC samples were produced by pooling 50 urine samples. These QC samples monitored the stability of our GC-MS over time by measuring intra-study reproducibility and allowed re-calibration of operation settings between batches, if needed. Moreover, they can be used to provide extra data for metabolite detection and analysis in future research. Our QC samples showed the precision and intermediate repeatability of our data.

Aggio et al. suggest freeze-drying as an effective alternative to simple treatment techniques, but it is not applicable to everyone because of limited access to freeze-driers and the long sample preparation time (up to 17 h) [6]. Simple treatment techniques, such as the addition of acids, require little training and are extremely easy. Moreover, Aggio et al. found freeze-drying to be comparable to HCl [6], whereas we have found H_2_SO_4_ to be significantly better than HCl. The ability to maximise the number and quality of VOCs produced from a sample will offer significant advantages to the field of metabolomics.

Future work will involve the application of this optimised method for the use in HS-SPME-GC-MS analysis to enhance our understanding of human disease and identify novel diagnostic markers of disease.

## 4. Materials and Methods 

### 4.1. Donor Recruitment and Ethical Consent

49 donors were recruited. Regulatory approval was granted by North Wales Research Ethics Committee–West (REC reference 15/WA/0464, REC name: Wales REC 5, Date of REC opinion: 21 December 2015). Written informed consent was obtained prior to recruitment.

### 4.2. Urine Samples

5 mL of urine was collected from each donor and frozen at −20 °C for storage. The collected urine was thawed at room temperature for 1–3 h, vortexed, and then divided into 1 mL aliquots contained within 10 mL vials. The new 1 mL urine aliquots were then refrozen and stored at −20 °C prior to sample preparation.

### 4.3. Chemicals and Materials

5 M sulphuric acid solution (H_2_SO_4_) and 5 M hydrochloric acid solution (HCl) were purchased from Fisher Scientific (Loughborough, UK). 5 M sodium hydroxide (NaOH) solution was obtained from Sigma Aldrich (Dorset, UK). The 10 mL vials (screw top, rounded bottom, clear glass vial) and caps (screw cap, magnetic, PTFE/silicone septum, septum thickness 1.3 mm) were also from Sigma Aldrich. The 2 mL vials (crimp top, clear. Vial size: 12 × 32 mm (11 mm cap)) and caps (crimp steel gold, magnetic, white silicone/PTFE septa, 11 mm) were purchased from Agilent Technologies (Cheshire, UK).

### 4.4. Experimental Conditions

We defrosted samples at room temperature prior to sample preparation. The following sample preparation experiments were performed:

#### 4.4.1. Altering Sample pH (and Ionic Strength)

Twenty-six triplicate urine samples of 1 mL were combined with 0.2 mL of 5 M H_2_SO_4_, 5 M HCl or 5 M NaOH (or saturated NaCl solution). The resulting concentration of H_2_SO_4_, HCl or NaOH treated urine sample solution was 0.83 M, pH 0.075, pH 0.081, and pH 13.919, respectively.

#### 4.4.2. Altering Concentration of H_2_SO_4_

Fifteen duplicate urine samples of 1 mL were combined with 0.2 mL of 5 M H_2_SO_4_, 2.5 M H_2_SO_4_ or 1 M H_2_SO_4_. The resulting concentrations of the urine sample solutions were 0.83 M, 0.42 M or 0.17 M respectively pH 0.075, pH 0.365, and pH 0.743, respectively.

#### 4.4.3. Altering Phase Ratio by Changing Vial Size

Fifteen duplicate urine samples of 1 mL were aliquoted into 0.5 mL urine samples and separated into 2 mL or 10 mL vials, respectively, and were combined with 0.1 mL of 5 M H_2_SO_4_ solution. The resulting concentration of each urine sample solution was 0.83 M, pH 0.075.

#### 4.4.4. Altering Phase Ratio by Changing the Sample Volume

Fifteen duplicate urine samples of 1 mL were chosen. One set of duplicates was treated with 0.2 mL of 5 M H_2_SO_4_ solution. The other set had 0.5 mL of urine aliquoted in a 10 mL vial with 0.1 mL 5 M H_2_SO_4_ solution. The resulting concentration of each urine sample solution was 0.83 M, pH 0.075.

All samples were vortexed for 10 s prior to analysis.

### 4.5. Static Headspace-SPME-GC-MS Analysis

A Perkin Elmer Clarus 500 GC-MS quadruple bench top system (Beaconsfield, UK) was used in combination with a Combi PAL PAL COMBI-xt autosampler (CTC Analytics, Switzerland, https://www.palsystem.com/). The GC column used was a Zebron ZB-624 with inner diameter 0.25 mm, length 60 m, and film thickness 1.4 μm (Phenomenex, Macclesfield, UK). The carrier gas used was helium of 99.996% purity (BOC, Sheffield, UK). A DVB/CAR/PDMS SPME fibre (needle size 23 ga, StableFlex, for use with autosampler) was obtained from Sigma-Aldrich (#57298U) and pre-conditioned before use, as per manufacturer’s instructions, 4 × 30 min heating at 300 °C. A blank was always then run to check the fibre was clean before use.

Urine samples were placed in a Combi PAL PAL COMBI-xt autosampler (CTC Analytics, Switzerland, https://www.palsystem.com/) incubation chamber at 60 °C for 30 min, followed by the extraction of volatiles from the headspace of the vial and adsorption to the SPME fibre for 20 min (static HS-SPME). The fibre was then inserted into the GC component for desorption at 220 °C for 5 min. The initial temperature of the GC oven was set at 40 °C, held for 2 min, increased to 220 °C at a rate of 5 °C/min and then held for 4 min, with a total run time of 42 min. A solvent delay was set for the first 4 min and the MS was operated in positive electron impact ionization EI+ mode, scanning from ion mass fragments 10–300 *m*/*z*, with an interscan delay of 0.1 s and a resolution of 1000 at FWHM (Full Width at Half Maximum). The helium gas flow rate was set at 1 mL/min. All samples were randomly ordered (using Microsoft Excel’s = RAND() function) and injected, essentially as described previously by Aggio et al. [6].

### 4.6. Library Building

After HS-SPME-GC-MS, chromatograms were generated (Figure 7, Table 2). Individual peaks were analysed by Automated Mass Spectral Deconvolution and Identification System (AMDIS, version 2.71, 2012, AMDIS is a freely available and sophisticated software for GC-MS data interpretation from NIST www.amdis.net) software, in conjunction with the National Institute of Standards and Technology mass spectral library (NIST, version 17, 2019, Purchased from SS Scientific Limited, Eastbourne, UK). Peaks were added to a local library: if they had a forward match of greater than 740/1000, they were silica based degradation products (5 compounds), if they had a good shape total ion count trace (2 unknown compounds), or if they had previously been identified in other studies in our lab. The final local library contained 173 unique VOCs (Appendix A).

A batch report was generated from AMDIS using the local library with deconvolution settings as follows: component width of 10, adjacent peak subtraction of one, low resolution, low sensitivity, and low shape requirements. Freely available R package Metab (Bioconductor) was used to generate a spreadsheet of VOCs per sample, using a half a minute time-window [55]. Contaminants identified with system suitability tests were removed before statistical analysis as described above. Statistics and figures were produced using Excel, the freely-available MetaboAnalyst website (McGill University, Canada, Xia Lab, https://www.metaboanalyst.ca/), or free software environment for statistical computing and graphics R software [56,57].

Data was normalized by median, transformed logarithmically, and auto scaled accordingly to ensure that all metabolites were given equal weight during the analysis and to increase the interpretability of the models produced.

### 4.7. System Suitability and Quality Control

A set of QC samples was produced by pooling 1 mL from 50 urine samples (including urine samples used in this paper) and aliquoting this into 1 mL samples. These acted as technical replicates run with each batch of study samples to provide a reference to indicate temporal stability measure HS-SPME-GC-MS. Parts of the SPME fibre or stationary phase of the GC column contain polysiloxanes that can degrade to produce volatile siloxanes [6]. Therefore, before every batch of sample processing, a “laboratory air” (uncapped, empty vial) was run, such that the SPME fibre sampled the air for 20 min at room temperature. We ran “blank” samples (capped, empty vials) after every 8 study samples. Compounds appearing in either laboratory air or blank samples revealed impurities and were considered contaminants if they were present in >50% of controls and blanks. These were removed from statistical analysis to ensure that analysed VOCs originated from individual urine samples being tested and prevented carry-over of VOCs on the SPME fibre between samples [5]. Collectively, QC, blanks, and laboratory air samples made up a set of system suitability samples, that were processed under the same experimental conditions as ‘real’ samples, and assured analytic metrics were “fit for purpose” [52].

### 4.8. Equipment Stability of the HS-SPME-GC-MS over Time

We assessed the clustering of QC samples using principal component analysis (PCA) plots. Additionally, the coefficient of variation (CV) was calculated per sample per treatment technique for each compound identified.

The acid treatment techniques, using H_2_SO_4_ or HCl, produced tightly clustered QC technical replicates that reflect the stability of the HS-SPME-GC-MS over the duration of the study. Importantly, this reflects the stability of the SPME fibre, despite the use of strong, oxidizing acid treatment techniques, and strengthens the robustness of our study. The technical replicate of NaOH treated urine also produced tightly clustered data in two distinct groups (Figure 6). As these were QC technical replicates, the intra-cluster variations must be due to an analytical variation in HS-SPME-GC-MS processing. Since both NaOH clusters remained within the 95% confidence interval of the study samples, this indicates the analytic variation was only minor and suggests that the quality of is good. The CV of VOC peak area was calculated for VOCs shared between QC technical replicates of each treatment technique. CV values were 0.07–4.33%, 0.15–5.28%, and 0.01–3.81% for H_2_SO_4_, HCl, and NaOH, respectively. Full lists are shown in Appendix A.

### 4.9. Compound Identification

Group arithmetic means were compared using a Wilcoxon-signed rank test. The presence of VOCs in each condition was compared using Venn diagrams generated through R software [58]. Univariate methods were used to measure differences in abundance: fold change (FC > 1.5, FC < −1.5) and *p* < 0.05 (false discovery rate [FDR] adjusted) were considered significant. The identified compounds were then grouped into chemical classes based on the MeSH (Medical Subject Headings) compound database [59].

### 4.10. HS-SPME-GC-MS Degradation

In order to identify the detrimental effect of each treatment on the equipment; we compared the abundance of contaminating metabolites originating from HS-SPME-GC-MS. The Wilcoxon-signed rank test was used to identify significant differences in HS-SPME-GC-MS degradation.

## 5. Conclusions

We optimised the detection of VOCs in urine using HS-SPME-GC-MS by altering pH, H_2_SO_4_ concentration, and phase ratio. This is the first direct comparison of H_2_SO_4_, HCl, and NaOH as treatment techniques prior to HS-SPME-GC-MS analysis. Altering urine sample pH indicates that H_2_SO_4_ is more effective at optimising detection of VOCs than NaOH or HCl. H_2_SO_4_ resulted in a significantly larger mean number of VOCs being identified per sample and more unique VOCs, produced more VOCs with significantly greater abundance than their counterparts, produced a more diverse range of classes of VOCs, and led to less HS-SPME-GC-MS degradation. We propose that adding 0.2 mL of 2.5 M H_2_SO_4_ to 1 mL of urine within a 10 mL headspace vial is the optimal sample preparation prior to HS-SPME-GC-MS analysis. We hope the use of our optimised method for urinary HS-SPME-GC-MS analysis will enhance our understanding of human disease and bolster metabolic biomarker identification.

## Figures and Tables

**Figure 1 metabolites-10-00482-f001:**
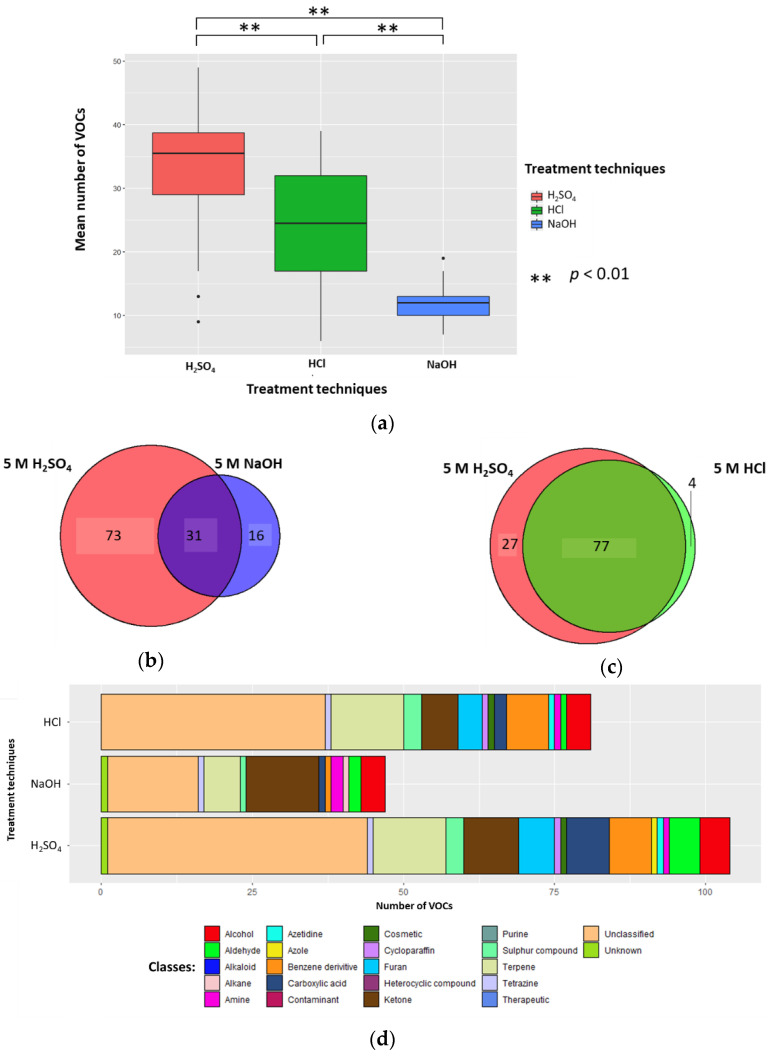
The urinary volatile organic compound (VOC) profiles after treatment with 5 M H_2_SO_4_ (red), 5 M NaOH (blue), or 5 M HCl (green) in the samples (to generate sample solutions of 0.83 M; pH 0.075, pH 0.081, and pH 13.919, respectively) (*n* = 26). (**a**) Boxplot to show the number of VOCs produced per treatment technique. (**b**) Venn to show the cumulative number of unique VOCs produced via addition of H_2_SO_4_ and NaOH. (**c**) Venn to show the number of VOCs produced via addition of H_2_SO_4_ and HCl. (**d**) Chemical classes of compounds produced when H_2_SO_4_, NaOH, and HCl were used as treatment techniques.

**Figure 2 metabolites-10-00482-f002:**
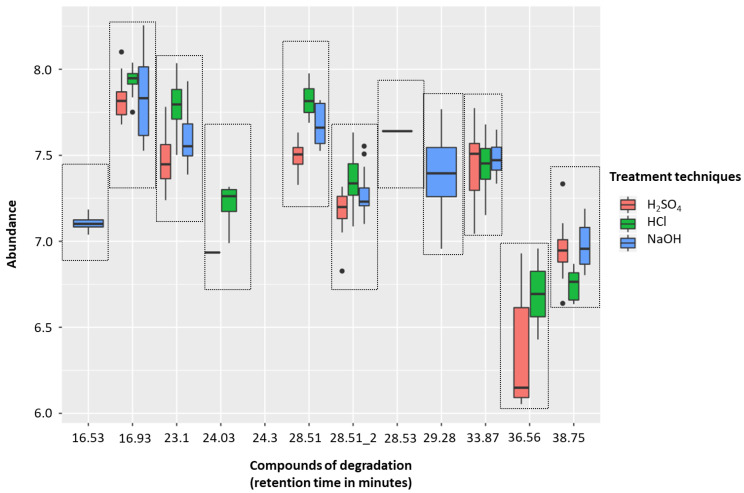
Headspace-solid phase microextraction gas chromatography-mass spectrometry (HS-SPME-GC-MS) degradation across 5 M H_2_SO_4_ (red), 5 M HCl (green), and 5 M NaOH (blue) treatment techniques (to generate sample solutions of 0.83 M; pH 0.075, pH 0.081, and pH 13.919, respectively). Box plots show the abundances of identified contaminants per treatment technique. Compounds of degradation are based on their retention times.

**Figure 3 metabolites-10-00482-f003:**
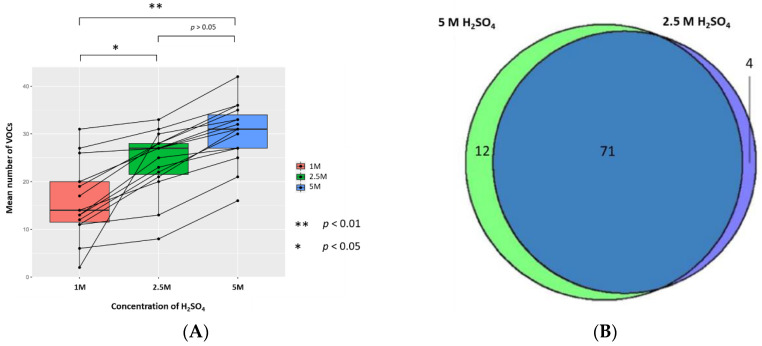
The urinary VOC profiles using 1 M (red), 2.5 M (green), and 5 M of H_2_SO_4_ (blue) as the treatment technique (to generate sample solutions of 0.17 M, 0.42 M, 0.83 M, pH 0.743, pH 0.365, and pH 0.075 respectively) (*n* = 15). (**A**) Paired boxplot to show the cumulative number of unique VOCs produced per sample for each concentration of H_2_SO_4_. (**B**) Venn diagram showing the unique VOCs detected between 5 M and 2.5 M H_2_SO_4_.

**Figure 4 metabolites-10-00482-f004:**
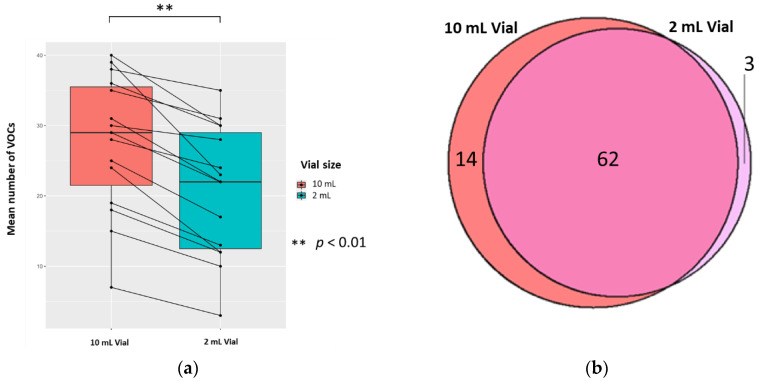
The urinary VOC profiles using 2 mL (red) and 10 mL (blue) vials (*n* = 15). 0.5 mL urine was treated with 0.1 mL 5 M H_2_SO_4_, to yield 0.83 M final solution, pH 0.075. (**a**) Paired boxplot to show the cumulative unique number of VOCs produced per sample for either vial volume. (**b**) Venn diagram showing the unique VOCs detected in each vial volume.

**Figure 5 metabolites-10-00482-f005:**
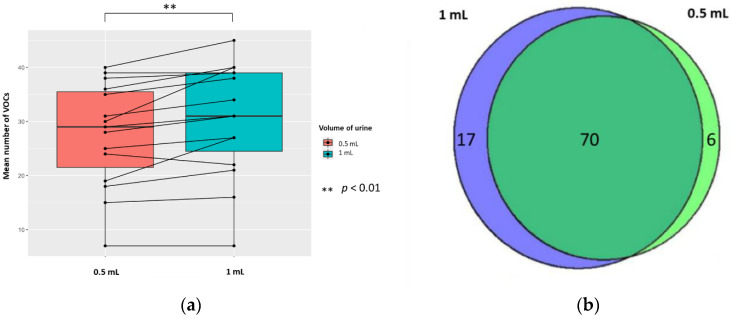
The urinary VOC profiles using 0.5 mL (red) and 1 mL of urine (blue) (*n* = 15) treated with 5 M H_2_SO_4_, to yield 0.83 M final solution, pH 0.075. (**a**) Paired boxplot to show the cumulative number of unique VOCs produced per sample for either volume of urine. (**b**) Venn diagram showing the unique VOCs detected in each volume of urine.

**Figure 6 metabolites-10-00482-f006:**
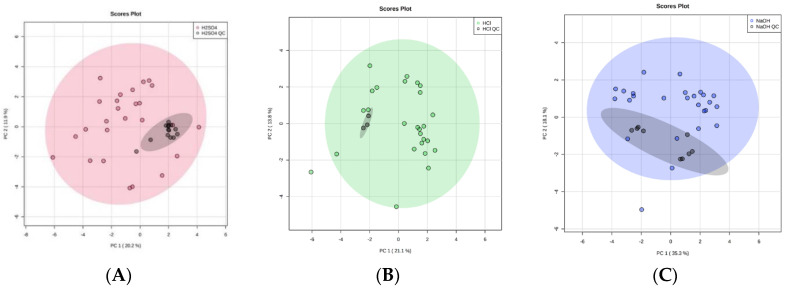
Principal component analysis (PCA) plots to illustrate equipment stability of the HS-SPME-GC-MS over time by comparing 1 mL ‘real’ urine samples with 1 mL QC technical replicates (black). With the addition of 0.2 mL of (**A**) 5 M H_2_SO_4_ (red), (**B**) 5 M HCl (green), and (**C**) 5 M NaOH (blue). 95% confidence intervals are displayed by the shaded ellipses: (**A**) 5 M H_2_SO_4_ (red), (**B**) 5 M HCl (green), and (**C**) 5 M NaOH (blue), and QC technical replicates (grey).

**Figure 7 metabolites-10-00482-f007:**
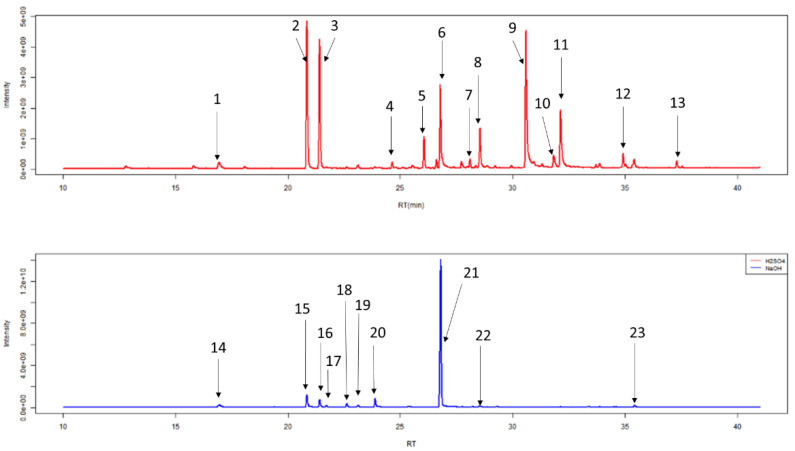
Example of typical chromatograms of duplicate 1 mL urine samples generated from HS-SPME-GC-MS following treatment with 0.2 mL of 5 M H_2_SO_4_ (red) or 5 M NaOH (blue). The 13 most abundant chemicals in urine treated with 5 M H_2_SO_4_ and nine most abundant chemicals in urine treated with 5 M NaOH are labelled and further information regarding these peaks can be seen in Table 2.

**Table 1 metabolites-10-00482-t001:** HS-SPME-GC-MS degradation relating retention time to its IUPAC name.

Retention Time (min)	Compound Name (IUPAC Name)	H_2_SO_4_ v HCl ^1^	H_2_SO_4_ v NaOH ^1^
16.53	dihydroxy(dimethyl)silane	-	<0.01
16.93	2,2,4,4,6,6-hexamethyl-1,3,5,2,4,6-trioxatrisilinane	<0.01	>0.01
23.1	2,2,4,4,6,6,8,8-octamethyl-1,3,5,7,2,4,6,8-tetraoxatetrasilocane	<0.01	>0.01
24.03	[(dimethyl-3-silanyl)oxy-dimethylsilyl]oxy-dimethylsilicon	<0.01	-
24.3	trimethoxy(methyl)silane	-	-
28.51	2-Hydroxymandelic acid, ethyl ester, di-TMS	>0.01	>0.01
28.51_2	2,4-bis(trimethylsilyloxy)benzaldehyde	<0.01	<0.01
28.53	Tetramethylsilane	<0.01	<0.01
29.28	2,2,4,4,6,6-hexamethyl-1,3,5,2,4,6-trioxatrisilinane	-	<0.01
33.87	Tetramethylsilane	>0.01	>0.01
36.56	trimethyl(1-trimethylsilylethyl)silane	>0.01	-
38.75	bis[[dimethyl(trimethylsilyloxy)silyl]oxy]-dimethylsilane	<0.01	<0.01

^1^ Wilcoxon-signed rank test (*p-value*).

**Table 2 metabolites-10-00482-t002:** Most abundant chemicals in chromatogram.

Label	Retention Time (min)	Compound Name (IUPAC Name)
1	16.93	2,2,4,4,6,6-hexamethyl-1,3,5,2,4,6-trioxatrisilinane (Contaminant) *
2	20.88	heptan-4-one *
3	21.44	heptan-3-one *
4	24.67	2-methyl-5-methylsulfanylfuran
5	26.01	1-methyl-4-propan-2-ylbenzene (*p*-Cimene)
6	26.93	2-ethylhexan-1-ol *
7	27.76	Phenol
8	28.55	1-methyl-4-prop-1-en-2-ylbenzene-or-1-methyl-2-prop-1-en-2-ylbenzene
9	30.61	4-methylphenol
10	31.83	(1R,2R,5R)-5-methyl-2-propan-2-ylcyclohexan-1-ol
11	32.14	5-methyl-2-propan-2-ylcyclohexan-1-ol(Menthol)
12	35.42	3-methyl-6-propan-2-ylcyclohex-2-en-1-one *
13	37.32	2-buta-1,3-dienyl-1,3,5-trimethylbenzene
14	16.93	2,2,4,4,6,6-hexamethyl-1,3,5,2,4,6-trioxatrisilinane (Contaminant) *
15	20.88	heptan-4-one *
16	21.44	heptan-3-one *
17	21.71	heptan-2-one
18	22.61	(E)-2-methylhept-2-enal
19	23.10	2,2,4,4,6,6,8,8-octamethyl-1,3,5,7,2,4,6,8-tetraoxatetrasilocane (Contaminant) *
20	23.91	Oxime-, methoxy-phenyl-(Contaminant)
21	26.93	2-ethylhexan-1-ol *
22	28.53	Tetramethylsilane (TMS) (Contaminant)
23	35.42	3-methyl-6-propan-2-ylcyclohex-2-en-1-one *

* Some compounds appear in this table twice as they are highly abundant in both treatment conditions.

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
