# Peer review of "Optimisation of Urine Sample Preparation for Headspace-Solid Phase Microextraction Gas Chromatography-Mass Spectrometry: Altering Sample pH, Sulphuric Acid Concentration and Phase Ratio"

_metabolites, 2020, doi:10.3390/metabo10120482_

Round 1

Reviewer 1 Report

This paper investigated the effect of altering urine sample pH, volume, and vial size for optimizing the detection of VOCs when using HS-SPME-GC-MS. The authors proposed to add 0.2 mL of 2.5 M H2SO4 to 1 mL of urine within a 10 mL headspace vial as the optimal sample preparation before HS-SPME-GC-MS analysis. HS-SPME-GC-MS analysis of urine potentially can offer for fast, non-invasive metabolite detection. The reviewer found that the work result could help develop a standardized protocol for the preparation of urine samples using HS-SPME-GC-MS. Some minor typos should be fixed before publication. Examples are listed below:

P1L26 “H2SO4”.

P2L76 “[8, 24].[6,27,28]” citations can be merged.

P3L96 “H2SO4”.

P13L387 “H2SO4”

Reviewer 2 Report

Aggarwal et al. describe the process of optimizing a headspace-solid phase microextraction gas chromatography – electron ionization- mass spectrometry based method for the analysis of urine samples. For that purpose, different parameters were tested: acidification via a strong acid using either HCl or H2SO4 or alkalization using the strong base NaOH. As a result, different concentrations of sulphuric acid were tested. The third parameter optimized was the employed vial size and sample volume.

The purpose of the paper, namely to provide an optimized method for the analysis of urine samples, would be clearly of interest to many readers. However, due to the use of this strong oxidizing acid, long term stability of the employed fibers will be most probably affected and hence robustness of this method needs to be further evaluated.

H2SO4 is a strong oxidizing acid, the use of this leads to the formation of sulfate in the gas phase and hence will most probably lead to deterioration of the fiber. Please provide a long-term stability study of the fiber in order to prove robustness of the optimized method.

Does an increase in the number of VOCs necessarily lead to high extraction efficiency or wouldn’t it be also possible that the use of this strong oxidizing acid, 5 M H2SO4, lead to an increase in decomposition/ degradation of certain compounds present in the sample? Please elaborate.

Did the authors test the optimized method also for human urine containing protein (proteinuria)? What effect does this change in matrix have?

How does the ionic strength come into play when comparing HCl, H2SO4 and NaOH? Please elaborate.

Why wasn't the ionic strenght increased by also adding salt, e.g. NaCl? Please discuss.

How were compounds associated to column degradation differentiated from contaminants stemming from septa (e.g. vial as well as GC inlet)?

Since emphasis is put on changing the sample pH, please state the respective values/ elaborate on this.

L86: Please state also information on the concentration (and add pH of sample) here.

L94: Why was a FC of 1.5 chosen instead of the commonly employed value of 2?

2.3.1 What sample volume was employed for this experiment?

L230: Please elaborate on this hypothesis (including references).

4.4.1./ 4.4.2 please state pH of the sample

4.5. Was static headspace-SPME-GC-MS analysis employed? Please state accordingly.

4.5 Please state the flow rate.

4.5 How long was the employed extraction time?

4.5 How many analyses could be performed per fiber?

L305: How was preconditioning performed?

L315: Please describe briefly how samples were randomized.

4.6 Did the authors consider making also use of retention indices for identification purposes? Please state also a level of confidence for the identified VOCs.

The authors state that hits were added to a local library if there was a forward match greater than 800- however in the supplementary file 3 values of <800 are also present.

Minor remarks:

L333: On what basis were data normalized? What scaling approach was applied?

L338: On what basis was the subset created? What were the selection criteria?

L76: Please correct: ”and or”

L88, L135: Please state also relative standard deviations.

Fig 1d: It is very hard to distinguish the different colors.

Please state already somewhere at the beginning of the results part the number of replicates employed for the different optimization steps.

L239: It would be helpful for the reader to know already that this state how the QCs were prepared.

Figure 7: It is very hard to differentiate between sample and QC in these figures.

Author Response

We have responded to all your comments in red.
We hope you find this a satisfactory response.

Reviewer 3 Report

The manuscript is devoted to the optimization of urine sample preparation for further headspace-solid phase microextraction GC-MS analysis with the VOC detection. The main idea of the manuscript is to check acidic and alkaline conditions, sample and vial volumes for better detection of larger number of VOCs. The most important findings of the work are: 1) sulphuric acid in the 2.5M concentration is the most appropriate for analyzing larger number of volatile organic compounds; 2) the vial volume is also important? And the optimal volume is 10 mL; 3) the appropriate sample volume is 0.2 mL. This work is an attempt to standardize the protocol of urine sample preparation for VOC analysis.

The manuscript is written in clear, concise style, and is easy to follow. The experimental design is well thought out, and the analysis is performed at modern level. I have just a few comments and questions:

  1. It is suitable to add to the introduction the explanation, which classes of VOCs are usually measure to identify different diseases or some kind of information will it be useful for MDs, for example, to know the concentrations or ratio of those VOCs that you can measure with your technique.
  2. Is it possible to apply this sample preparation for whole blood, plasma or serum?
  3. May be I missed somewhere, but how did you identify the exact compounds? If you can identify each compound on Figure 6 with NIST library, it is better to add annotation to the chromatogram.
  4. Figure2 and Table1 – the sign X with RT confused me a bit. What does it mean?
  5. Line 130 – in two times.
  6. As a remark – it is better to place Figure 1 to one side, otherwise there is a mess with a-b-c-d letters.

Summarizing, the manuscript is definitely worth publishing in Metabolites.

Reviewer 4 Report

Moreover, urine is often used in experiments because it is non-invasively collected from patients. The manuscript doesn't show new technique of urine samples preparation. It would much more interesiting if any markers of diseases were pointed out in such study. This is much more interesting in the aspects of metabolites analysis.

Also some minor mistakes were found out in the text.

No numerical fidings in abstract part.

Keywors should be different than title. 

Introduction should be writen as a one part.

Reference no 26. should be Drabińska, N., Starowicz, M., Krupa-Kozak, U.

Round 2

Reviewer 2 Report

None of these two major points were properly addressed in the revised version.

* the use of a strong oxidizing acid, that is 5M H2SO4,will lead to two severe effects:
- first, the use of a strong oxidizing acid will change the sample and hence bias the results. Adding an oxidizing reagent to a metabolomics sample will lead to a change of the metabolic profile (both forms of a metabolite will be then present, part of the reduced form and part of the oxidized form(s))
- second, it will destroy the fibre over time and long-term stability of these (rather expensive fibres) will degrade over time and this will also affect the sensitivity - the authors neither answered my question on long-term stability nor give any indication on how many times the fibres can be re-used if 5 M H2SO4 is employed.

Author Response

We thank the reviewer for their valuable critique and comments.

  1. We acknowledged that strong oxidizing acid will change the sample composition in discussion paragraph 2: Different urine sample treatment techniques alter the sample matrix. Firstly, change in pH may lead to an increase in decomposition or degradation of certain compounds present in the sample.”.

In this latest version, we have gone further and added “In particular, using strong oxidizing acid will almost certainly change the sample composition.” (line 214-215).

  1. We know the fibres are stable for at least 15 days, sampling over 200 samples using 5M H2SO4. Further work is needed to examine longer time frames/periods.

Previously added in the 3rd paragraph of the discussion ‘Both acids may lead to some deterioration of the SPME fibre.’, we have now added, “this requires further study” (line 229-230)

We previously, added an entire paragraph about the potential problem of fibre degradation in our discussion – 5th paragraph. Where we explicitly say that future studies should also include a study of the long-term stability of the fibre under the acidic conditions. We have now elaborated further on this and added “where we ran over 200 samples,” (line 255), and “These further studies on SPME fibre degradation are vital, since SPME fibres have a high cost associated with them, therefore it is important to consider how many samples can be run per fibre to make this a viable avenue moving forwards.“ (line 261 – 264).

We analysed 234 urine samples over 15 days treated with 5M H2SO4 (without loss of performance). Each fibre used in this study, #57298U from SIGMA (we have added this product code to line 357 on the manuscript), cost approximately £100 GBP therefore using 5M H2SO4 for 200 samples, costs approximately 50p per sample. Further work is needed to examine increased sample numbers. We believe at 50p per sample the use of 5M H2SO4 is not cost prohibitive.

Reviewer 4 Report

I don't see the novelty of this manuscript, therefore I don't recommend to published in Metabolites with high IF. This manuscript could be an introduction part in an article where some biomarkers of the disease will be presented. Urine is mostly used because it can be collected non-invasively, also there are much better publications with biomarkers already available. 

Author Response

  • We thank the reviewer for their comments. We believe the data looking at sulphuric acid is novel, and the ensuring comparison with sodium hydroxide and hydrochloric acid are thus also novel. The findings are unexpected and will interest other researchers that look a biological sample, especially urine.
  • This work is an important foundation for the wider work of our GC-MS lab, and we hope our findings will be of benefit to the wider scientific community that use GC-MS (to analyse urine). The in-depth method development in this paper underpins future/imminent clinical research publications from our lab.
